# Identifying Opportunities for Irrigation Systems to Meet the Specific Needs of Farmers in East Africa

Georgia D. Van de Zande *, Susan Amrose, Elliott Donlon, Pulkit Shamshery and Amos G. Winter V

Department of Mechanical Engineering, Massachusetts Institute of Technology, Cambridge, MA 02139, USA; samrose@mit.edu (S.A.); edonlon@mit.edu (E.D.); p.shamshery@gmail.com (P.S.); awinter@mit.edu (A.G.W.V.)
* Correspondence: gdvdz@mit.edu

**Abstract:** East Africa (EA), a region facing food shortages, has very little irrigation adoption compared to the rest of the world. Increasing irrigation has been shown to increase food cultivation, so governments and private organizations have been attempting to introduce irrigation products into the EA market. Despite this support, irrigation adoption rates remain low, reflecting that existing solutions do not meet the needs of medium-to-small-scale farmers. Meeting these needs is challenging due to the diversity of farmers in EA and the minimal exploration of these differences in the literature. This study sought to elucidate some of these differences and explore whether new opportunities exist for irrigation products targeting EA farmers. An interview-based market assessment was first conducted to identify key market segments and unique values that farmers in each segment may hold for an irrigation system for each segment. Then, a techno-economic feasibility analysis was used to reveal which combinations of irrigation methods and energy sources present promising opportunities for each segment. Four distinct market segments were found. Broadly, the traditional smallholder would likely most value a system that uses photovoltaic (PV) power and manual irrigation. The semi-commercial smallholder may find promising opportunities in a system that uses PV power and butterfly sprinklers. Both the medium-scale contract farmer and the remote farm owner would likely value PV panel- and drip irrigation-based systems. These identified opportunities can guide innovation for irrigation designers as they create new systems to directly serve the needs of specific market segments, with the aim of increasing irrigation and food security in EA.

**Keywords:** drip irrigation; East Africa; farmer-led irrigation; irrigation; irrigation markets; manual irrigation; photovoltaic-powered irrigation; sprinkler irrigation; water–energy–food nexus

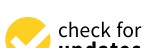



## 1. Introduction

The objective of this study was to identify new opportunities for innovation to enhance the adoption of farmer-led irrigation systems in East Africa (EA) by elucidating and targeting the needs of distinct market segments. This study specifically sought to understand whether new opportunities exist for sustainable irrigation products targeted at small-to-medium-scale farms. In 2020, an estimated 39.4% of EA's population was food insecure; this number will likely still remain above 20% in 2030 [1]. Increasing irrigation is an effective path toward increasing food cultivation [2,3]; however, this is a farming practice that is not widely adopted in EA. Only 2.2% of the cultivated land in EA is currently irrigated, compared to 22.7% in North Africa, 39.1% in Asia, and 19.7% worldwide [4,5]. Farmer-led irrigation on small-to-medium-scale farms—as opposed to large public schemes or large commercial farms—has been suggested as a promising path to increase irrigation in EA [6]. Governments, non-governmental organizations (NGOs), and private companies have been looking for solutions to increase farmer-led irrigation to meet food demands, proposing diverse solutions such as treadle pumps, drip irrigation kits, and various motorized pumps [7]. However, despite this institutional support, no single solution or set of solutions has been able to significantly increase EA irrigation adoption.

The lack of irrigation adoption suggests that existing irrigation products do not meet the cost and performance requirements for much of the market. Small- and medium-scale farmers in EA (cultivating 5 ha or less) account for 95% of Sub-Saharan African farm holdings [8], but they are not all similar. One challenge in developing irrigation solutions is the high degree of diversity among EA farmers [9,10]. These farmers have a wide range of irrigation needs, differing in their typical crops, irrigation schedules, farm area, and expected outcome of irrigation (e.g., subsistence or business growth). Different market segments would likely respond better to irrigation solutions that deliver value propositions targeted to their specific needs [11]. The literature focuses largely on smallholders who cultivate less than 2 ha [3,7,12–17]. These farmers have many limitations such as low ability to pay [18], with few opportunities to afford irrigation systems with their desired level of performance. There are also a large number of farms in the 2–5 ha range that may not be well served because their cost and performance requirements are not well understood. To address these challenges, this work investigated the needs of unique market segments and proposes opportunities for the design of irrigation systems targeting those unique needs.

In addition to the need for increased food security in EA, there is also a strong need to accomplish this goal sustainably to avoid calamities that have occurred in other regions when irrigation coverage was increased. Expanded irrigation during the Green Revolutions in China and India increased food security but also severely depleted each country's water resources [19,20]. Africa may be going through a similar revolution [21,22], so it is important to consider how water-saving technologies or emission-free energy sources can be introduced while still meeting the unique needs of farmers in different market segments. This study therefore considered irrigation methods and energy sources that are water-efficient and emission-free.

This study addressed the following research questions:

- What distinct market segments exist within the range of small-to-medium-scale EA farmers who participate in farmer-led irrigation? What are the user-driven needs of farmers in each of these segments?
- How do these unique needs translate into value propositions for irrigation systems that can articulate pathways to achieve the most desired irrigation benefits within the constraints of each segment?
- What technical requirements can be discerned from the user needs and value propositions of each market segment? How do those requirements compare to the current performance of feasible systems within each context?

These questions were explored in a two-part analysis. An interview-based market assessment was first used to segment the range of small-to-medium-scale farmers into distinct user groups and elicit farmers' needs and corresponding value propositions. A first-order techno-economic analysis was then used to explore which technologies could be feasibly realized energetically and then propose sets of requirements for irrigation products that could deliver the desired value to farmers. The outcome of this work highlights potential opportunities for technical innovation that could increase the likelihood of user-driven adoption of farmer-led irrigation in EA.

## 2. Interview-Based Market Assessment

### 2.1. Interviews with Farmers and Key Market Stakeholders

Qualitative interviews with farmers and key market stakeholders were used to gather information about the current irrigation market and farmers' typical irrigation preferences, challenges, constraints, and agricultural goals. Throughout 2019–2021, 34 semi-structured interviews with farmers were conducted. Interviews had a typical duration of 30 to 60 min and guided subjects through questions about likes and dislikes of their current irrigation systems, noticeable improvements of their current system over any previous irrigation methods, typical irrigation schedules, household and agricultural water usage,

well installation processes (if applicable), ability to repay their current systems, willingness and ability to pay for new equipment, and future plans for improving their farms.

Interview subjects were selected to cover a range of both field size and level of irrigation experience, including subjects who irrigated small vegetable gardens using buckets to those managing flower export businesses using drip irrigation in greenhouses. To target lead users and early adopters who are known to provide useful feedback at this early stage of any design process [23], only subjects with prior irrigation experience were considered. In total, 23 subjects were recruited in Kenya (one of whom was interviewed twice), 4 in Ethiopia, and 6 in Zambia (Figure 1). Although Zambia is not in EA, the recruited Zambian farmers were successful adopters of farmer-led irrigation schemes that were similar to those in Kenya, enabling them to provide useful insights about potential future opportunities. Kenyan farmers were most heavily recruited because EA farmer-led irrigation innovations frequently begin in Kenya before moving to neighboring countries. Twenty-five interviews were conducted on subjects' farms, which allowed for the inclusion of photos and observations of farm conditions as further qualitative data. Twenty-four farmers were recruited through private irrigation companies or NGOs, including SunCulture, Futurepump, iDE, Water4, Inc., and Illumina Africa. Several of these interviews were conducted through the use of a local translator. Due to international travel restrictions in 2020 and 2021, the remaining nine interviews were conducted remotely by phone. Remotely interviewed farmers were recruited through an online survey that was promoted by several irrigation equipment suppliers. All interview protocols were approved by the Massachusetts Institute of Technology Committee on the Use of Humans as Experimental Subjects.

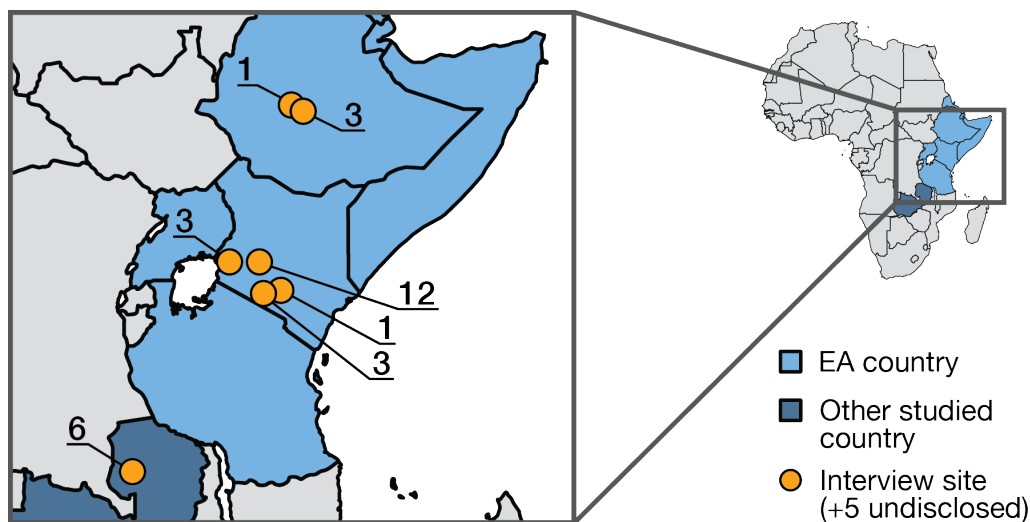

**Figure 1.** EA countries, locations of interview sites, and the number of interviews conducted at each site. For five remote interviews, subjects did not disclose their specific locations within Kenya. As noted, Zambia is not in EA, but these interviewed farmers had similar irrigation success as EA farmers, so these interviews were included in the study.

To understand the irrigation products currently offered in the EA irrigation market and explore the envisioned future of the market, 47 semi-structured interviews were conducted with key, non-farmer stakeholders throughout 2019–2021. Stakeholders were recruited who were knowledgeable in the preferences and constraints of small-to-medium-scale farmers from a range of diverse perspectives. These stakeholders represented government agencies, NGOs, irrigation equipment distributors, agricultural input suppliers, borehole drilling companies, agricultural research organizations, agricultural universities, and microfinance institutions (MFIs). Twenty-one stakeholders were based in Kenya, thirteen in Ethiopia, and the rest were based outside of EA but had relevant expertise by serving or studying farmers in this region. Each type of stakeholder was asked about how their region has changed in recent years with respect to improving irrigation, what ideal irrigation would

look like from their perspective, and what it might take to reach this ideal. Four subjects were interviewed more than once to collect additional follow-up information. Data from these 30- to 60-min-long interviews complemented the farmer interviews by providing insight into the current state of the irrigation market from a broader policy and technology perspective. This complementary approach is visualized in Figure 2.

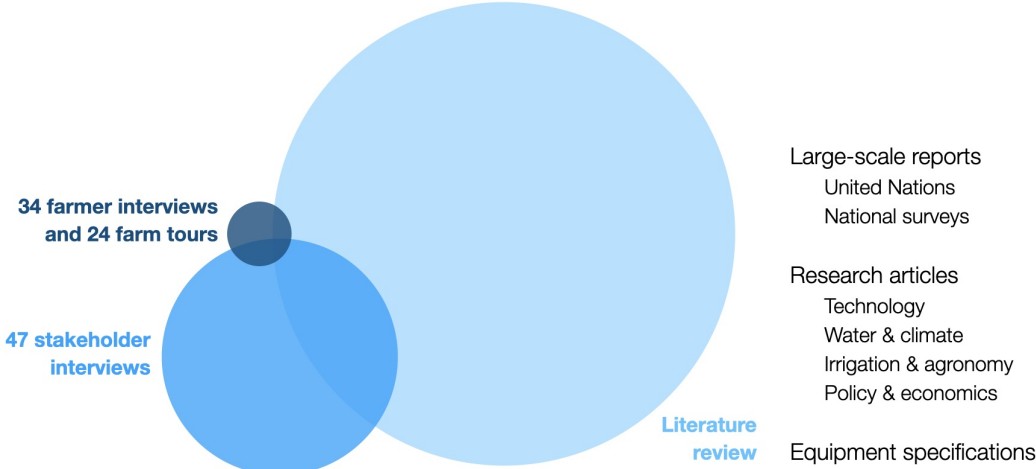

**Figure 2.** Visualization of the broad perspective built through a combination of farmer interviews, farm tours, stakeholder interviews, and literature review. Interviews with farmers provided the richest data about farmers' values but were limited in number. Stakeholders who served or studied thousands of farmers provided a broader perspective, but their insights were further removed from direct farmer experiences. Literature—including large-scale reports, research articles, and equipment data sheets—provided insights based on large numbers (e.g., millions) of farmers but with less detail about farmer experiences. These three levels of insight were combined to build both a deep and broad perspective of the studied farmers.

### 2.2. Market Segmentation Based on User-Driven Irrigation Needs

As hypothesized, the irrigation needs and contexts of the interviewed farmers were highly diverse, as summarized in Table 1. Farmers had access to a variety of different water sources, irrigated a wide range of farm areas, and grew diverse crops. Interviewed farmers spent varying amounts of time in the fields and utilized different irrigation schedules. They had different financial constraints and expectations with respect to the lifetime and lifetime cost of a system. Farmers had previously purchased irrigation systems, indicating their willingness and ability to pay for irrigation equipment. These systems had differing performances, prices, and payment plans. These variations in responses were examined to identify clusters of similar farmer traits, which were synthesized into unique market segments. Farmers were grouped together based on the following factors:

- Their farms were similarly sized and they were located in similar regions (e.g., rural or peri-urban);
- They cultivated the same types of crops (e.g., fruits, vegetables, or grains) and they used those crops for the same reasons (e.g., for in-home consumption or for market selling);
- They had similar economic profiles in terms of willingness and ability to invest in irrigation equipment;
- Their current irrigation practices were similar in terms of experience on similar equipment and similar irrigation knowledge or training, irrigation scheduling, and maintenance;
- Their desire for additional value add-ons that an irrigation system could offer were similar (e.g., cell phone charging or solutions for remote farm management).

Segmentation yielded four distinct market segments:

- The traditional smallholder (10 farmers in this segment were interviewed);

- The semi-commercial smallholder (14 farmers were interviewed);
- The medium-scale contract farmer (7 farmers were interviewed);
- The remote farm owner (2 owners were interviewed).

**Table 1.** Summary of user-driven irrigation needs and value propositions for the four market segments discovered.

| | Traditional Smallholder | Semi-Commercial Smallholder | Medium-Scale Contract Farmer | Remote Farm Owner |
|---|---|---|---|---|
| Water source(s) | Surface water or shallow wells up to 10 m deep | Surface or shallow wells/ boreholes up to 25 m deep | Boreholes up to 100 m deep | Boreholes up to 100 m deep |
| Farm area irrigated | 0.125 ha | 0.25 ha | 2–5 ha (this analysis uses 4 ha) | 1–4 ha (this analysis uses 2 ha) |
| Irrigation scheduling | Willing to irrigate 4 h/day | Willing to irrigate 6 h/day | Willing to irrigate 7 h/day | Willing to irrigate 7 h/day |
| practices | Can be flexible with crop subsections as needed | Can be flexible with crop subsections as needed | Crop subsections are $\geq$0.2 ha and take $\geq$0.5 h to irrigate | Crop subsections are $\geq$0.2 ha and take $\geq$0.5 h to irrigate |
| Crop types and intended use for crop | A mix of low-value crops (e.g., maize) and high-value vegetables (e.g., cabbage and tomato) Intend to consume >90% of crop yield | High-value vegetables (e.g., cabbage, tomato) and fruits Intend to consume >70% of crop yield and sell >30% | High-value crops (e.g., tomatoes, cabbage, herbs, fruits) Intend to sell >95% of crop yield | High-value crops (e.g., tomatoes, cabbage, herbs, fruits) Intend to sell >100% of crop yield |
| Investment timescale | 2–3 seasons (this analysis uses 1 year) | 2–3 years (this analysis uses 3 years) | 5–10 years (this analysis uses 5 years) | 5–10 years (this analysis uses 5 years) |
| Lifetime cost over investment timescale | USD 300; USD 200 before value add-ons | USD 1300; USD 1000 before value add-ons | USD 18,000; USD 15,000 before value add-ons | USD 9000; USD 7500 before value add-ons |
| Non-irrigation value add-ons | Phone charging and home lighting; Under 50 kg for portability | Phone charging, home lighting, power for small home appliances (e.g., TV, fan, minifridge, and cooking appliances) | Increased data and prediction (irrigation, pests, disease, markets) Flexibility of system based on farm characteristics | Solutions for improved remote farm management Same value add-ons as medium-scale contract farmer |
| Core value proposition of an irrigation system | A low-cost, portable irrigation system that replaces human power and enables cell phone charging and home lighting | An irrigation system that helps farmers grow their businesses and lifestyles | An irrigation system that maximizes farmers' profits | An irrigation system that farmers can monitor from the city and provides them with additional income |

These results were synthesized to generate hypotheses about the irrigation needs of farmers in the different segments. The hypotheses were then validated with data from stakeholder interviews based on their broad knowledge base and from the literature review (Figure 2). These hypotheses produced a series of qualitative farmer profiles that suggest the user-driven irrigation needs of each market segment. User profiles—also known as personas—are a common design tool that generalize the attributes of a group of people into one set of characteristics, even though individuals in the group exhibit a range of characteristics. Such profiles are useful tools in early-stage design processes because they help designers focus on a set of users and the users' goals, rather than focusing on technological limitations [24]. They also help ensure that solutions will work well for the group, rather than the individual.

Sections 2.2.1–2.2.4 present the resulting farmer profiles, elaborating on the needs summarized in Table 1. Appendix A provides further elaboration on these profiles, including details about farmers' irrigation experiences, socio-economic statuses, geographic locations, and motivations for investing in irrigation. Additional citations relating to farmer profiles are also provided in Appendix A.

### 2.2.1. Profile of the Traditional Smallholder

The traditional smallholder is a subsistence farmer cultivating on a small, rural plot (average 0.125 ha), whose main farming motivation is to grow food for their families. The vast majority of traditional smallholders have minimal or no irrigation experience. Of those who do irrigate, most rely on manual irrigation. Attitudes towards manually powered pumps with low capital costs, such as the treadle pump, revealed both the high value placed on low-cost irrigation and the high physical toll of supplying the water manually. Attitudes toward risk and income generation patterns suggest that traditional smallholders tend to be highly risk-averse and would value a system that could be paid off in 2–3 seasons. The risk aversion of traditional smallholders also leads them to diversify their crop selections (including both grains and vegetables) and their income sources, using this as a way to mitigate risk. Traditional smallholders are thus less willing to invest all their time and money in farming activities. These farmers also value obtaining more functionality from their irrigation system hardware than irrigation alone. For example, increasingly more farmers in this segment have home lighting and cell phones that are powered from the same source as their irrigation system.

### 2.2.2. Profile of the Semi-Commercial Smallholder

The semi-commercial smallholder was likely a traditional smallholder at some time in the past. Now, they have moved away from subsistence farming to attempt to start a small farming business. Compared to the traditional smallholder, they are more willing to invest both time and money in equipment, particularly if it has a promising return on investment because they have some experience with successes in agriculture. Compared to traditional smallholders who have diverse income sources, farmers in this market segment are more focused on farming as their main income source. Therefore, they are able to dedicate more time per day to irrigation than traditional smallholders. Semi-commercial smallholders grow largely the same types of crops as traditional smallholders, with slightly more focus on higher-value fruits and vegetables over grains. While still located in rural areas, semi-commercial smallholders are quick to implement new agriculture techniques once they have access to the right resources. Like traditional smallholders, farmers in this segment are also interested in system capabilities beyond just irrigation. Interview results suggested they could derive value from small home appliances, like televisions and pressure cookers, and that they are willing to pay for these extra items.

### 2.2.3. Profile of the Medium-Scale Contract Farmer

Medium-scale contract farmers run full-time farming businesses to feed the growing cities in EA. They cultivate medium-sized farms (typically 2–5 ha) in peri-urban areas. Farmers in this market segment invest heavily in their businesses. Intending to sell >95% of their produce, they cultivate high-value crops like tomatoes, herbs, and fruit. Medium-scale contract farmers have advanced irrigation experience compared to smallholders. They employ seasonal and full-time laborers who irrigate, weed, plant, and harvest. Because farmers have this additional help, they are willing to spend a larger portion of their day irrigating. These farmers focus on selling their produce, so the appearance and size uniformity of their crop is very important.

### 2.2.4. Profile of the Remote Farm Owner

The remote farm owner lives in a city but owns or rents additional land in a nearby peri-urban region. They farm as a hobby or as a way to make supplemental income while investing in the land. The remote farm owner is not typically present on the farm on a daily basis, though they are likely involved in making big decisions about the farm. They tend to hire farm managers and laborers to run the farm for them day-to-day. The remote farming market segment is still emerging and many challenges of remote farm management have not been solved, resulting in increased risk for the owners. Interviewed farm owners cited instances where hired laborers claimed to have completed work that was not actually done.

Farmers in this segment likely have the capital to invest in irrigation systems, but they do not intend for farming to be their main income source and so may rather invest their capital elsewhere.

### 2.3. Value Propositions of Irrigation Systems for Each Market Segment

The farmer profiles were used to gain insights into the design constraints and irrigation system performance requirements that are most significant or highly valued by farmers in each market segment. These insights were used to build value propositions for irrigation systems that could fulfill the needs of farmers in each market segment (Table 1). These value propositions are discussed further in Sections 2.3.1–2.3.4.

#### 2.3.1. Traditional Smallholder

The value proposition of an irrigation system designed for a traditional smallholder is a low-cost, portable irrigation system that replaces human power and enables cell phone charging and home lighting.

First, the system must replace human power to provide value and ease the heavy burden felt by manual irrigation. Many EA farmers have leveraged human-powered irrigation to come out of poverty [25]; however, this is extremely hard and labor-intensive work. A system that replaces the human as the energy source could allow farmers to significantly improve their quality of life. Further, with the introduction of a system that replaces human power, farmers could shift their efforts to other income-generating tasks. Assuming they would want to spend about half their time on other income-generating activities, a system that only requires a farmer's attention for only 4 h/day would satisfy this need.

Second, a system for the traditional smallholder must be portable. Stakeholders noted that theft of irrigation equipment is a major issue for smallholders, so farmers must be able to bring the equipment inside their homes each night. Smallholders do not necessarily cultivate on plots of land nearby their homes, so they need equipment that is easily transported, weighing a maximum of 50 kg [26].

Third, traditional smallholders value a system that enables cell phone charging and home lighting in addition to irrigation. The growing number of traditional smallholders with cellphones and access to home lighting also suggests that these farmers may value more than just the irrigation ability of a system [27,28]. Bundling in-home lighting and phone-charging capabilities with an irrigation system may provide additional value to farmers that promotes adoption. Benchmarking comparisons to current products in the local market that offer these capabilities suggest that a system should provide power for three home lights in the evenings and the daily charge of two cell phones in addition to fulfilling irrigation needs.

Fourth, the system must be low-cost, with a USD 300 target cost paid out over three seasons. Traditional smallholders' high risk aversion suggests they are unlikely to invest in an agriculture product that is not guaranteed to benefit them in a short time scale (on the order of 2–3 seasons). KickStart International has concluded that USD 200 is the target capital cost for a system that fills a similar set of irrigation needs for this market [26]. However, KickStart's proposed system does not provide the additional cell phone charging and home lighting value. Adding these features (valued at USD 100 by interviewed farmers) to KickStart's estimated USD 200, we estimated a lifetime target cost of USD 300 would best serve this market.

Finally, the system must fulfill the irrigation needs of the typical crop selections of the traditional smallholder. Subsistence farmers grow a variety of crops to feed their families, ranging from low-value crops like maize to higher-value vegetables like cabbage. Given this range of crops, cabbage was selected as a representative crop because it captures the higher end of what a traditional smallholder might expect in terms of water demand. Because they are primarily growing for their families and have small land holdings, they only need to irrigate approximately 0.125 ha. The water sources available to these farmers

are surface water and shallow wells or boreholes up to 10 m deep. Deeper sources are neglected because they would be too expensive for a traditional smallholder to install without the support of the government or an NGO.

### 2.3.2. Semi-Commercial Smallholder

The value proposition of an irrigation system designed for semi-commercial smallholders is a system that helps them grow their businesses and lifestyles.

First, the system must meet the farmer's changing business needs. Because semi-commercial smallholders are growing businesses, they are good candidates for a system that allows the user to incorporate add-ons or switch out components to improve irrigation performance over time. Their ability and willingness to learn new farming techniques further reinforces the value of this feature.

Second, the system must accommodate a rapidly changing lifestyle. The system should have at minimum the capability to light a home and charge a cell phone, as seen with the traditional smallholder. The system should further be able to power small home appliances as the farmer is able to afford them, including a television, cooking appliances, a chaff cutter, an egg incubator, fans, or a minifridge (all examples given in interviews).

Third, the system must meet the irrigation needs of a semi-commercial smallholder. Based on interviews, it was found that farmers are willing to irrigate their 0.25 ha of land for up to 6 h/day on average. Shallow groundwater up to 20 m deep is an accessible, strategic resource for many smallholders throughout EA [29]. However, it is noted that existing products serving a small percentage of this market operate at slightly deeper depths. In particular, SunCulture's RainMaker2 with ClimateSmart™ Battery is a photovoltaic (PV)-powered irrigation system that is designed to operate best at 32 m pressure head [30]. Aiming between these two values, this analysis proposes 25 m as a representative water source depth for this market segment. This segment tends to sell higher-value crops, so tomatoes are used as a representative vegetable.

Finally, the system must meet the tight budget constraints of the semi-commercial smallholder. In total, 13 out of 14 interviewed farmers owned Futurepump or SunCulture PV-powered irrigation systems. Depending on the configuration, these systems cost USD 600–1550 and are paid for over 2–3 years. Using these systems as benchmarks of viable systems in this market segment, this analysis proposes a target cost of a novel system at USD 1300, paid over three years. Our analysis suggested that farmers require higher flow rates than these benchmark systems. However, USD 1000 (the average cost of SunCulture and Futurepump systems) is suggested for the irrigation system alone, based on the success of similarly priced products for existing farmers in this market segment. Adding approximately USD 300 worth of add-on features (e.g., appliances) brings the total target cost to USD 1300. While some successful products exist for some semi-commercial smallholders, many farmers in this segment have not yet adopted one of these products, suggesting that the current offerings do not yet meet their needs.

### 2.3.3. Medium-Scale Contract Farmer

The value proposition of an irrigation system designed for medium-scale contract farmers is a system that maximizes their profits.

Such a system can maximize a farmer's profits in two ways: by minimizing expenses or by maximizing revenue. A system can minimize expenses by decreasing operating costs, decreasing capital costs, and decreasing labor needs. Selecting an appropriate irrigation strategy helps farmers decrease their operating and capital costs, which this work aims to address.

Labor costs can be decreased by introducing automation on a farm. According to an interviewed system designer, large-scale farmers (who are not examined in this work) typically use a high degree of automation on their fields. However, this technology is out of reach for many medium-scale farmers due to its high expense. One farmer remarked that he is ready to automate his irrigation, but a company quoted him USD 38,000 for a fully

automated system for his approximately 1 ha farm. Another farmer who recently installed a USD 30,000 irrigation system did not yet have automation, but said he would consider purchasing it if it were about 10% of the total system cost. The interviews demonstrated a need for more affordable automation that is accessible to the medium-scale contract farming market.

A system can maximize revenue in several ways, for example by providing increased data and prediction tools, which can increase farm yields. Several current smartphone apps aim to help farmers understand market trends so they can make educated decisions about harvesting [31,32]. In the Australian market, research is being performed to help farmers predict their yields and increase farm profits [33]. A system that can help farmers plan for unexpected weather, market, disease, or pest trends could provide great value to this market segment by influencing how farms are managed.

The irrigation system must always also meet the irrigation needs of the specific market segment. Within the scope of this work, we propose a representative farm that is 4 ha, growing high-value crops, and irrigated with water from a 100 m deep borehole. To estimate the borewell depth, we obtained company logs for 2470 borewells drilled by Hydro Water Well(K)Ltd. (Nairobi, Kenya) in EA over the past 23 years. The most common borehole depths were between 100 and 125 m. These wells serve more than just medium-scale contract farmers, but the managing director agreed that many of his customers fitting this profile had boreholes about 100 m deep.

The system must be flexible to accommodate the diverse farm characteristics found within this market segment. Although this work proposes a representative 4 ha farm and 100 m borewell depth, our data suggest that medium-scale contract farms cover a wide range of conditions. Within this segment, some farmers used a surface water source to irrigate 5 ha, while others used a 200 m deep well to irrigate 2 ha. Some equipment, like drip irrigation lines or PV panels, is highly modular and can be readily scaled to match farm characteristics. Other components, such as pumps, may require a suite of options to cover the market conditions. To accommodate this flexibility, a central system controller that can offer high performance in a variety of changing conditions could provide great value.

Finally, the system must deliver a high value for the monetary investment. Based on data gathered about the cost of existing medium-scale contract systems (see Appendix A), this analysis proposes an estimated lifetime system cost of USD 18,000, paid over five years. Because farms in this segment have a large range of sizes and water source depths, this target value is expected to have a high level of variation, with USD 18,000 used for the representative case outlined in Table 1. Excluding any sensors or a controller that would provide additional value, this cost lowers to an estimated target of USD 15,000 for just the irrigation components of the system. However, the most significant constraint is clearly demonstrating to the farmer that the proposed investment would increase their profit.

### 2.3.4. Remote Farm Owner

The value proposition of an irrigation system designed for a remote farm owner is a system that farmers can monitor from the city and that provides them with additional income.

First and foremost, the irrigation system for the remote farm owner must be profitable because it functions primarily as an investment. Farm parameters in this segment are similar to medium-scale contract farm parameters because both focus on profit as the main motivation. Remote farm owners typically hire managers with similar levels of agricultural experience as medium-scale contract farmers. However, the size of the irrigated area depends on the capital that a remote farm owner is willing to invest. They may not have as much direct experience as a contract farmer and so many not have seen past successes in agriculture the way a medium-scale contract farmer might have. Thus we propose a smaller 2 ha irrigated area as representative. The target costs are similarly scaled by a factor of 0.5, setting the target of the entire system at USD 9000, with the irrigation components alone at USD 7500.

Second, the system must allow for remote monitoring. Available products and services do not yet fully support the needs of the remote farm owner in this respect. A system that could track weather, soil moisture, fertilizer application, and irrigation activity would provide great value and a sense of confidence to this market segment. However, an irrigation system alone might not fulfill all the remote farm owner's needs. Interviewed subjects in this segment noted that it was extremely difficult to manage and trust the laborers they had hired to manage the farm. These challenges suggest that a professional farm management service could be valuable for remote farm owners.

### 3. Techno-Economic Feasibility Analysis

*3.1. Estimating the Cost of an Irrigation System*

The farmer profiles and value propositions outlined in Section 2 provide quantitative performance requirements that must be met to satisfy user needs in each market segment (Table 1). To estimate the cost of delivering this performance, a first-order techno-economic model was built. This model was used to assess common and emerging irrigation strategies. For this analysis, an "irrigation strategy" is a combination of an energy or power source plus an irrigation method (e.g., grid electricity + sprinkler irrigation). The resulting analysis incorporates the user-driven irrigation needs, data on the cost and performance of different irrigation strategies, and pump cost estimations.

For each irrigation strategy, system operating points (flow rate $Q$ [m$^3$/h] and total dynamic head $h_{tot}$ [m]) were calculated using

$$Q = 10 \left( W_c \, A_{field} \, f_w \right) / t_{irr} \tag{1}$$

and

$$h_{tot} = h_{water} + h_{equip}, \tag{2}$$

where $W_c$ is the daily crop water requirement [mm], $A_{field}$ is the field's irrigated area [ha] from Table 1, $f_w$ is a unitless water factor specific to the irrigation equipment's water usage efficiency, $t_{irr}$ is the daily irrigation time [h] from Table 1, $h_{water}$ is the head of the borehole or well [m] from Table 1, and $h_{equip}$ is the head of the irrigation equipment [m]. The water factor describes how much water a specific irrigation method uses. It is a unitless ratio of the volume of water used by the irrigation equipment over the volume of water needed by rainfall to produce the same crop yield. Irrigation methods with low water factors save water without sacrificing crop yield. The values from Table 1 are held constant for the analysis of each market segment. The daily irrigation time is held constant at its maximum value to minimize the irrigation system cost that falls within the constraints of the market segment. If farmers irrigate the maximum number of hours they are available in a day, a result found in Section 2, an irrigation system's flow rate is minimized for a given irrigation volume demand (another result found in Section 2). As flow rate decreases, power and pumping costs decrease, minimizing overall system cost.

Submersible multistage centrifugal pumps are suitable to use in boreholes and wells commonly found in EA. For each of the cases, a suitable pump with the best efficiency point closest to the farm's operating point ($Q$ and $h_{tot}$) was chosen from Alibaba.com's online catalog. Stakeholders cited this catalog as a source for the low-cost pumps commonly found in EA. The pumps were primarily selected from two vendors: Hangzhou Qinjie Electromechanical Co., Ltd. (Hangzhou, China), which offers low-power DC solar pumps [34], and Taizhou Qingquan Pump Co., Ltd. (Taizhou, China), which offers higher-power AC pumps [35]. The pump efficiency $\eta_{pump}$, pump price $C_{pump}$ (assumed equal to the single unit price listed on Alibaba.com), and pump lifetime (assumed equal to the pump warranty) were incorporated into the system cost estimations based on the available pump specifications. The pump pricing was based on the manufacturers' high volume (>50 pieces) listing prices, excluding shipping fees. The efficiencies of the selected pumps were obtained from the manufacturer's efficiency testing data.

For PV-powered systems, the power $P$ [W] required to reach the desired operating point was calculated using

$$P = \frac{\rho_w g Q h_{tot}}{3600 \eta_{pump}},$$ (3)

where $\rho_w$ is the density of water and $g$ is the acceleration due to gravity.

For systems using grid electricity or fuel, the daily energy $E_{daily}$ [MJ] required to meet the irrigation demand was calculated using

$$E_{daily} = \frac{\rho_w g Q h_{tot} t_{irr}}{3600 \eta_{pump}}.$$ (4)

The systems' capital cost $C_{cap}$ [USD] was estimated using pump costs, irrigation equipment costs, and power costs (if applicable) using

$$C_{cap} = C_{pump} + A_{field} C_{equip\ per\ area} + P C_{Watt}.$$ (5)

The irrigation equipment cost includes the upfront cost to the farmer of equipment necessary to carry out the irrigation strategy, such as hoses, field pipes, sprinklers, or drip lines. It excludes the cost of installation, training, water source access, and pipes from the water source to the pump.

The systems' operating costs $C_{op}$ [USD] were estimated using applicable energy costs and

$$C_{op} = t_{eval} \left( 365 E_{daily} C_{MJ} + \sum_{component} \left( \frac{C_{rep.}}{LT_{equip}} \right) \right),$$ (6)

where $t_{eval}$ is the evaluation time period set by the needs of each market segment, $C_{rep.}$ is any component replacement costs (equal to their capital costs), and $LT_{equip}$ is the corresponding expected lifetime of the equipment. The operating cost includes the cost of electricity or fuel to transport water from the source to the crops, but excludes the cost of hired labor.

This first-order analysis assumes that the available power is constant over the duration of the irrigation event, a valid assumption for fuel-based solutions. This assumption is less valid for grid- and PV-powered solutions due to potentially intermittent grid power and variable solar irradiance throughout the day, respectively. However, this first-order analysis provides an estimate of the order of magnitude cost for a given irrigation strategy. This first-order study excludes hired labor costs. Labor can be one of the larger operating costs for farmers, but these costs are highly variable and region-specific, and therefore outside the scope of this preliminary feasibility analysis. To compensate for this limitation, labor is evaluated qualitatively considering the needs of each market segment. Additionally, the costs of the tanks, batteries, filters, and fertigation units were neglected for these preliminary calculations because they do not vary significantly between the irrigation strategies considered here. Additionally, the possibility of different soil conditions was not considered in this analysis. Although soil type and texture can influence irrigation demand, they do so less than the variables considered in this first-order analysis, such as crop water requirement, field area, and water factor.

Some model inputs vary depending on the irrigation needs of different market segments, such as the area irrigated, the daily irrigation time, the depth of a borehole or well, and the crop water requirement (Table 1). Other model inputs are independent of market segment, such as the irrigation equipment cost per hectare, the equipment lifetime, the equipment operating pressure, the irrigation method water factor, and the cost of energy sources (Tables 2–4). Values for each of these inputs were gathered from the literature and interviews with distributors, who provided data on their current products. The source of each input is noted by citations in the respective table or by interview data and justifications presented in Appendix B.

**Table 2.** Crop water requirement parameters used as inputs for the techno-economic feasibility model [36].

| Irrigation Demand | Crop Water Requirement, $W_c$ |
|---|---|
| Medium (cabbage is representative) | 5 mm/day |
| High (tomato is representative) | 7 mm/day |

**Table 3.** Irrigation method parameters used as inputs for the techno-economic feasibility model.

| | Equipment Cost [USD/ha] $C_{equip\ per\ area},$ $C_{rep.}$ | Equipment Lifetime [years] $LT_{equip}$ | Operating Pressure [m] $h_{equip}$ | Water Factor $f_w$ |
|---|---|---|---|---|
| Manual irrigation | 50 | 2 | 1 | 0.5 |
| Flood or furrow irrigation | 25 | 2 | 1 | 1.0 |
| Butterfly sprinklers | 26.5 | 2 | 10 | 1.0 |
| NPC drip sections | 2400 | 3 | 14 | 0.5 |
| LE PC drip sections | 6000 | 10 | 5.9 | 0.5 |

**Table 4.** Energy source parameters used as inputs for the techno-economic feasibility model.

| | Cost, $C_{Watt}$, $C_{MJ}$ | Equipment Lifetime [years], $LT_{equip}$ |
|---|---|---|
| PV panels | 0.81 USD/W | 20 |
| Grid electricity | 0.06 USD/MJ | N/A |
| Fuel | 0.03 USD/MJ | N/A |

The irrigation methods and energy sources in Tables 3 and 4 were selected either because they are commonly used throughout EA or because they are emerging strategies that may not be widely used, but have the potential to create an impact in the region if introduced at scale. Combining data from interviews and literature, four irrigation methods, one emerging irrigation method, and three common energy sources were selected. The considered irrigation methods were as follows:

- Manual irrigation: Using buckets or handheld hoses to deliver water to the field;
- Flood or furrow irrigation: Covering the entire field with water or filling furrows between crop beds with water, respectively;
- Butterfly sprinklers: For this analysis, it is assumed that a farmer uses one set of five sprinklers that they move throughout their field every 30 to 60 min (based on interview data of common practices);
- Non-pressure-compensating (NPC) inline drip irrigation: Drip irrigation works by delivering water to rows of crops through a network of stationary main and submain pipes and lateral lines. The emitters within the lateral lines do not compensate for pressure changes expected in a pipe network, so the flow can be non-uniform;
- Low-energy pressure-compensating (LE PC) inline drip irrigation: PC drip emitters regulate their flow rates given the pressure changes expected in a pipe network, so flow is uniform throughout the field. LE PC drip is an emerging technology developed by the MIT Global Engineering and Research (GEAR) Lab [37]. LE PC emitters activate at lower pressures than conventional PC emitters, giving them the potential to save 42–54% in pumping power, which is an attribute that has shown promise in EA [38,39].

The considered energy sources were as follows:

- Photovoltaic (PV) panels;
- Grid electricity;
- Fuel (e.g., diesel or petrol).

Details of these irrigation methods and energy sources, as well as relevant citations and justifications for why they were selected, are noted in Appendix B.

### 3.2. Candidate Irrigation Systems for Each Market Segment and Their Estimated Costs

Irrigation methods and energy sources were assessed for their suitability to meet the needs of each market segment based on the segment's user-driven irrigation needs and value propositions from Table 1. An irrigation method or an energy source was assumed to be a candidate unless there was a user need or value that suggested it was a non-candidate. Table 5 presents the candidate and non-candidate irrigation methods and energy sources. In the case of a non-candidate method or source, a justification is given.

**Table 5.** Candidate and non-candidate irrigation methods and energy sources for each market segment, based on user-driven needs.

| | Traditional Smallholder | Semi-Commercial Smallholder | Medium-Scale Contract Farmer | Remote Farm Owner |
|---|---|---|---|---|
| Manual irrigation | Candidate | Non-candidate. The time needed to manually irrigate 0.25 ha for 6 h/day is too high for a farmer who is growing their business. | Non-candidate. The irrigated areas of 2–5 ha are too large for this irrigation method. | Non-candidate. The irrigated areas of 1–4 ha are too large for this method. |
| Flood or furrow irrigation | Candidate | Candidate | Non-candidate. Crop uniformity is important for resale value in this segment, and this method does not ensure uniformity. | Non-candidate. Crop uniformity is important for resale value; this method does not ensure uniformity. |
| Butterfly sprinkler irrigation | Candidate | Candidate | Non-candidate. The irrigated areas of 2–5 ha are too large for this irrigation method. | Non-candidate. The irrigated areas of 1–4 ha are too large for this method. |
| Drip irrigation (NPC or PC) | Non-candidate. Farmers lack the amount of training needed to use drip effectively. [1] | Candidate [2] | Candidate | Candidate |
| PV panels | Candidate | Candidate | Candidate | Candidate |
| Grid electricity | Non-candidate. Farms are too rural to have reliable connections. [3] | Non-candidate. Farms are too rural to have reliable connections. [3] | Candidate | Candidate |
| Fuel | Non-candidate. The high and fluctuating cost of fuel is a crutch in farmers' budgeting. Fuel can also be difficult to source. [4] | Non-candidate. The high and fluctuating cost of fuel is a crutch in farmers' budgeting. Fuel can also be difficult to source. [4] | Candidate | Candidate |

[1] One stakeholder who sells irrigation equipment in Ethiopia says that farmers need 5+ years of training before they can use drip effectively. Another stakeholder from an NGO said he has seen inexperienced farmers using drip lines as fencing or to tie up cattle. This happened because the farmers did not have the experience level necessary to use drip technology properly. [2] Unlike traditional smallholders, many semi-commercial smallholders have more access to professional training, so both NPC and LE PC drip irrigation are feasible methods for this market segment. [3] Farms for these two market segments are too rural to have reliable connections [40]. [4] Several interviewed farmers conveyed that the high and fluctuating cost of fuel created great difficulty in their monthly budgeting. While fuel is currently the most common energy source for the minority of smallholders who have graduated from manual or rainfed irrigation, given its observed drawbacks, this work assumes that fuel is not a viable energy source for the majority of traditional smallholders.

Using inputs from Tables 1–4, Equations (5) and (6) were used to estimate the system costs of candidate irrigation strategies. Figure 3 demonstrates the five candidate systems with the lowest lifetime cost and the corresponding estimated capital and operating cost for each market segment. For both smallholder markets, there were fewer than five candidate irrigation strategies, so fewer are displayed. In all four market segments, PV panel-based systems had the lowest lifetime cost, followed by fuel- and grid-based systems, respectively. In all segments, the PV panel- and LE PC drip-based systems had the highest capital cost. For medium-scale contract farmers and remote farmers who operate on a longer investment

timeline than smallholders, the LE PC drip-based systems had a lower lifetime cost than the NPC drip-based systems.

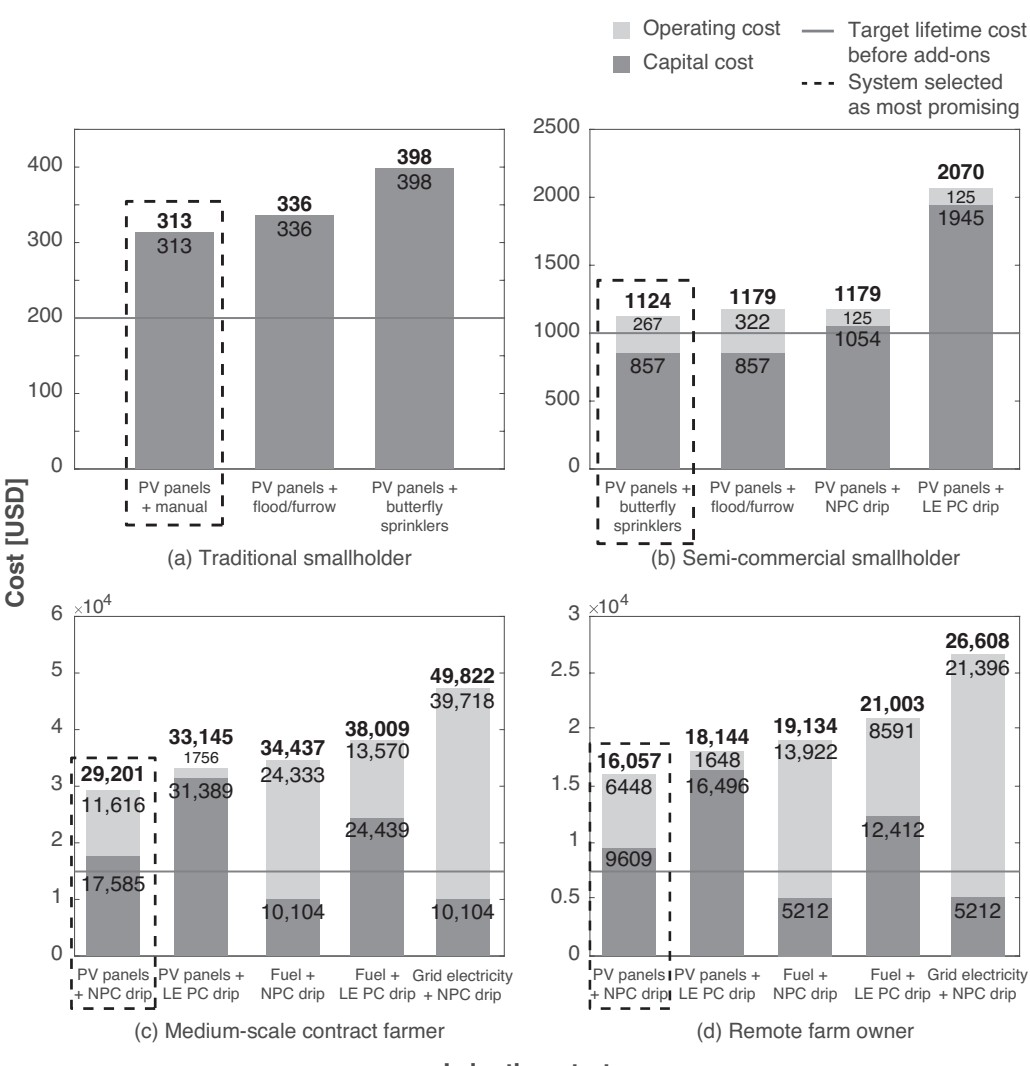

**Figure 3.** Estimated system lifetime, capital, and operating costs for candidate irrigation strategies, ranked by lowest lifetime cost for each market segment.

### 3.3. Discussion of Opportunities to Deliver on Value Propositions and Irrigation Needs

The most promising irrigation strategy was chosen for each market segment (boxed irrigation strategies in Figure 3), synthesizing results on farmer needs and values (Table 1), farmer risk adverseness (Sections 2.2.1–2.2.4), and estimated system costs (Figure 3). The most promising strategies were selected because they were most likely to deliver on the segment's value proposition and irrigation needs. On a case-by-case basis, the insights about user needs and value propositions were weighed against the results of estimated system costs. This method, detailed in the following paragraphs, factored in both user-driven requirements and technical limitations, allowing for a holistic assessment to select promising novel irrigation systems for EA farmers. The results suggest high-level design requirements for irrigation systems with the potential to deliver more value to farmers than existing systems (Table 6).

The system flow rate, system pressure head, and estimated costs were determined from the techno-economic feasibility analysis, described in Section 3. The maximum pump diameter is based on the water source appropriate for each segment and the system lifetime is based on the investment timescale constraints of each segment. The target selected

lifetime cost is repeated from Table 1. Because value add-ons (e.g., home lighting, cell phone charging, and small appliances) were not considered in the feasibility analysis, they are not repeated here, though they are important considerations for irrigation engineers to incorporate in system designs.

**Table 6.** Summary of most promising opportunities for irrigation systems and their corresponding technical requirements.

| | Traditional Smallholder | Semi-Commercial Smallholder | Medium-Scale Contract Farmer | Remote Farm Owner |
|---|---|---|---|---|
| Irrigation strategy | PV panels + manual irrigation | PV panels + butterfly sprinklers | PV panels + NPC drip irrigation | PV panels + NPC drip irrigation |
| System flow rate [$m^3$/h] | 0.8 | 2.9 | 20 | 10 |
| System pressure head [m] | 11 | 35 | 114 | 114 |
| Maximum pump diameter [cm] | 15 cm | 10 cm | 10 cm | 10 cm |
| Minimum lifetime [years] | 1 | 3 | 5 | 5 |
| Estimated system costs [USD] | Capital: 313 Operating: 0 Lifetime: 313 | Capital: 857 Operating: 267 Lifetime: 1124 | Capital: 17,585 Operating: 11,616 Lifetime: 29,201 | Capital: 9609 Operating: 6448 Lifetime: 16,057 |
| Target system lifetime costs [USD] | 300 USD; 200 USD before value add-ons | 1300 USD; 1000 USD before value add-ons | 18,000 USD; 15,000 USD before value add-ons | 9000 USD; 7500 USD before value add-ons |

From Table 6, none of the considered systems meet the low target lifetime cost of the traditional smallholder market segment. The most promising system is a combination of PV panels + manual irrigation. This USD 313 system has the lowest lifetime and capital costs of all considered candidate systems. PV panels were the only power source available to this market segment due to the lack of operating costs. There are no foreseen equipment replacements required within the 1-year planning horizon of the traditional smallholder. The selected irrigation strategy is water-efficient, increasing the sustainability of this solution. To meet the daily irrigation needs of this segment, the system would operate at 0.8 $m^3$/h at a pressure head of 11 m. Pumps must be <15 cm in diameter to fit in hand-dug wells.

For the three remaining market segments, pumps must be <10 cm in diameter to fit in common 4-inch boreholes. This was reported as the standard borehole diameter for EA by all interviewed stakeholders.

The most promising irrigation strategy for semi-commercial smallholders is PV panels + butterfly sprinklers. At USD 1124, this irrigation strategy produces a system with the lowest estimated lifetime cost. The estimated capital cost is equivalent to the PV panel + flood/furrow option, but the preferred strategy has a lower operating cost. No user-driven needs were proposed to justify selecting the flood/furrow system over the butterfly sprinkler system. Flood/furrow irrigation requires significant labor to prepare the field, whereas butterfly sprinklers require minimal but ongoing labor to move the sprinklers throughout the field each day. The PV panel + NPC drip system has a similar lifetime cost to the PV panel + sprinkler. However, this system has a 23% higher capital

cost, which is likely to turn off semi-commercial smallholders who are very sensitive to capital cost.

The most promising irrigation system for medium-scale contract farmers is PV panels + NPC drip irrigation. These farmers value the potential profit that a drip irrigation system could deliver and this irrigation strategy produces a system with the lowest estimated lifetime cost. While the fuel + NPC drip option is appealing due to its low capital cost (43% lower than PV panels + NPC drip), it has an 18% higher lifetime cost, impacting its profit potential. For medium-scale contract farmers who can afford the capital investment, the PV panels + NPC drip system is more suitable. As reported in Section 2, many farmers in this market segment have already experienced past successes in farming and are likely to have built some capital reserves in addition to being less risk-averse than smallholders. This suggests that the higher capital cost system with lower lifetime system cost based on PV panels + NPC drip irrigation is most suitable. To meet the average irrigation needs of this segment, the system would operate at 20 $m^3$/h and 114 m of pressure head. Such a system would also have to incorporate flexibility to accommodate the wide range of farm characteristics found in this market.

The remote farm owner would be best served by a PV panel + NPC drip irrigation system. The justification for this selection parallels that of the medium-scale contract farmer. Remote farm owners would further value the weather data that typically comes with a PV panel-based system, which could be remotely monitored. Owners in this segment tended to value more advanced technology, preferring PV panels over diesel or petrol as energy sources. PV panels also require less labor than a fuel-based system, alleviating some of the labor concerns of the remote farm owner. To meet the irrigation needs of this segment, the system would operate at a 10 $m^3$/h flow rate and 114 m pressure head.

## 4. Discussion

The results of this analysis demonstrate that the estimated best system cost is higher than the target cost for each of the four market segments considered (Table 6). This highlights the need for technological innovation to produce systems that are capable of meeting the needs of small- and medium-scale EA farmers. There are several potential strategies that could be used to lower system costs. Pumps that are longer lasting or less expensive could significantly reduce system costs. The analyzed pumps were sold with 1- or 2-year warranties, suggesting that a farmer might need to replace them several times within the lifetime of their other irrigation equipment. More expensive, longer-lasting pumps are available from manufacturers like Xylem and Grundfos, but farmers rarely select these pumps due to their high capital costs. These results demonstrate a need for innovation to produce longer-lasting pumps at lower prices or develop financial mechanisms and marketing schemes that would allow farmers to take advantage of existing options.

Another potential opportunity for technical innovation is in the design of irrigation systems. The first-order design conventions used to size the systems for this analysis do not consider nuances such as seasonal variation of solar irradiance, strategic hydraulic layouts to minimize pressure losses, local components to reduce costs, or strategic scheduling of irrigation to more efficiently utilize solar power. A systems-level optimization model that incorporates key farm parameters, local weather data, locally available system components, hydraulic optimization, or innovative control strategies could potentially help irrigation engineers design the systems identified in this work at a lower cost without sacrificing reliability. Such a design strategy could potentially help lower operating and capital costs and improve reliability because the system would not be over- or undersized. A design and optimization theory has recently been proposed to address this opportunity [41].

A third opportunity for innovation is in the creation of longer-lasting and less expensive drip irrigation equipment. Such innovation could benefit the EA irrigation market and promote a more sustainable expansion of irrigation to areas that are currently under-irrigated. LE PC drip equipment has a potential 10-year lifetime, but a high capital cost compared to NPC drip or traditional irrigation methods, making it inaccessible to many

farmers in EA. NPC drip equipment has a lower capital cost but a reduced 3-year lifetime. One possible opportunity to reduce drip system costs while increasing system lifetime is to modify the thickness of drip line tubing or its material properties. Based on interviews with drip manufacturing engineers, a large component of the product cost is due to the drip line material. This wall thickness is proportional to both the equipment cost and its lifetime, with thicker tubing costing more but lasting longer. Material innovations that would increase lifetime at a lower cost would add value to both medium-scale contract farmers and remote farm owners in EA.

A fourth opportunity for innovation is to increase drip line lifetime by designing better anti- or low-clogging drip emitters. Farmers and stakeholders claimed that clogging of emitters is a large drawback to the technology and the major reason for drip line replacement. Drip systems typically require filtration to prevent clogging, which increases capital cost and requires extensive maintenance. To reduce upfront costs, EA farmers typically install drip without any filtration, causing the emitters to clog more quickly. Farmers are not typically well-trained on how to care for their drip systems to prevent clogging. Stakeholders involved in distributing irrigation equipment have seen extensive drip disadoption due to emitter clogging. Innovation in this space could increase system lifetime and potentially reduce the training threshold required for effective drip use.

Further opportunities for innovation exist surrounding value add-ons, such as those presented in Table 1. For the traditional smallholder, creating an irrigation system that delivers more value than just irrigation, such as enabling phone charging and home lighting, can promote adoption. Technologies exist to combine these functions, but products have not yet penetrated this market. For the semi-commercial smallholder, an irrigation system that can also power small home appliances would be very valuable. SunCulture's ClimateSmart™ with Battery systems pair with some appliances, and in interviews, they reported that this significantly improved their sales and customer satisfaction. However, there is an opportunity to expand the available options, particularly for high-quality, low-cost DC appliances capable of directly pairing with a solar power system. For medium-scale contract farmers and remote farm owners, value is added when systems can be modular and highly flexible to accommodate different farm parameters and farm parameters that change over time. One potential opportunity to create flexibility is to design a central controller that can be paired with different equipment, such as pumps, which would enable operation at the most efficient operating point based on a particular farm's current characteristics. This controller could potentially integrate with low-cost sensors to provide a farmer with predictive insights on how to manage their farm.

One limitation of this study is that it does not investigate how governmental policies, institutional support, and infrastructural changes could impact irrigation adoption in EA. This study instead focused closely on the perspectives of farmers. If technical limitations are faced, policy or business innovation may be an alternative path to help system costs match performance needs. Government subsidies, sponsorship programs, or loan programs are some ways to increase the adoption of irrigation systems when farmers are unable or unwilling to pay the full price. Improved or expanded extension services could help teach farmers how to best utilize different systems to meet their needs within their constraints. Marketing campaigns to convey the potential value that these systems could provide could also increase the amount farmers are willing to pay for irrigation systems. Professional farm management services or other institutional supports could potentially help remote farm owners, particularly if remote monitoring of sensors does not provide them with the confidence they need to ensure quality labor when they are not on site. Conversely, policy and institutional barriers can also restrict technical innovation and limit access to irrigation equipment. A full perspective analysis of policy and institutional design levers and barriers will be an important consideration before implementing the strategies or solutions proposed here.

This exploratory work aimed to identify and substantiate potential opportunities for irrigation innovation that might improve adoption of farmer-led irrigation in EA. While some sustainability concerns were considered, it was outside the scope of this work to fully quantify and assess the environmental impacts and sustainability of the proposed irrigation solutions in EA. The environmental impact of different irrigation solutions can be complex. For example, while drip irrigation can reduce water consumption in many circumstances, some communities have associated drip adoption with increased groundwater consumption due to overwatering, expansion of irrigated area, and conversion to higher-value crops with a higher water consumption [42,43]. A complete assessment of the potential social impact of the solutions should consider these complexities.

Similarly, it was outside of the scope of the present study to assess the potential impact of different future scenarios or trends on the feasibility of proposed irrigation solutions. For example, solar panels are reducing in cost and increasing in availability across much of EA [44], which could change the outlook for solar-powered options. The present work acts as a starting point to identify innovation gaps and potential opportunities, while future work will explore more fully how these opportunities might change in the future.

This exploratory analysis focuses on differences among solutions that meet the irrigation system requirements of generalized market segments of farmers. A degree of variation exists and is expected within each segment across multiple metrics (e.g., willingness and ability to pay, daily irrigation demand, land availability, and access to agricultural training). This study explores solutions with the potential to satisfy a significant portion of key market segments but acknowledges that additional work is required to understand the full extent of variation within each segment as well as the full array of potential segments among small- and medium-scale farmers. Some farmers will diverge from the personas outlined in Sections 2.2.1–2.2.4. For example, they may be more or less willing and able to pay for a system with the target costs in Table 1. Irrigation equipment designers can use the technical specifications presented in this paper to understand key differences between the segments, create benchmarks, and identify segment-specific innovation targets.

It was outside of the scope of this analysis to consider combinations of irrigation strategies utilized by the same farmer. Combination strategies could be valuable to some farmers, particularly semi-commercial smallholders who value growing their businesses and lifestyles. For example, a farmer could adopt a sequential arrangement of strategies, starting with the PV panel + sprinkler system and then expanding to a new area with a drip system that utilizes the same pump, once they have accrued profit from the initial strategy. The 0.25 ha field of butterfly sprinklers used in this analysis requires a flow rate of 2.9 m$^3$/h and a pressure of 35 m while an expanded 0.5 ha field of NPC drip irrigation operates at the same flow rate and 39 m of head. A single pump could provide both flow rates with the addition of USD 216 of PV panels. The pump and PV panels comprise 99% of the original capital cost, which farmers could leverage upon expanding their field. Further analysis of this type of growth strategy could highlight additional opportunities for innovation.

The present study is limited by the relatively small number of farmer interviews conducted. A small number of in-depth interviews were chosen to gain a better understanding of nuances between the values of different market segments. Data from these interviews were supplemented with data from stakeholder interviews and the literature to offer a broader perspective (visualized in Figure 2). Additional interviews or survey data could strengthen confidence in the user-driven irrigation needs of Table 1 or provide more clarity on metrics such as willingness and ability to pay for irrigation systems. A larger number of subjects could also elucidate additional farmer segments or aspects of the present segments that were not observable with the sample size of the present study. Continued engagement with farmers and stakeholders during the innovation process will help refine and expand the conclusions of this study.

This study is also limited by the number of irrigation methods and energy sources considered. More than four irrigation methods and three energy sources exist and are used successfully in EA. Those selected were considered to be the most promising by the

majority of interviewed stakeholders and farmers. A larger analysis could incorporate additional methods and sources in future work.

An additional limitation of this work is the assumption that the lifetime of a pump is equal to its warranty period. Several farmers and stakeholders noted that a pump with a two-year warranty might last up to five years. However, equipment distributors typically provide little-to-no information about the expected product lifetime beyond the warranty period. In the case of the medium-scale contract farmer, equating the pump lifetime to the one-year warranty results in 16% of the estimated operating cost (USD 3528) being attributed to annual pump replacements. If a more reliable estimate of the lifetime could be found, this cost would likely decrease, placing the irrigation system costs closer to the target costs identified in Table 6.

A final limitation of this work stems from excluding the cost of farmers' and laborers' time from the system cost. Farmers with diversified income sources have an opportunity cost when they spend multiple hours per day on the farm. Farmers who hire laborers to operate their irrigation systems must pay these workers, adding to their operating costs. These costs are difficult to incorporate because they are highly variable and context-specific. However, these costs could influence which irrigation strategy appears most promising for a specific market segment. For example, if traditional smallholders have high opportunity costs, they may value a manual irrigation-based system less than the value assigned in this work. Instead, a flood- or furrow-based system could provide the most value. Remote farm owners and medium-scale contract farmers are most likely to pay significant labor costs. However, including labor costs is unlikely to change the conclusions of Table 6 because all of the systems considered in Figure 3c,d would have similar labor costs. In these cases, the operating costs for each strategy would be higher than estimated, potentially reducing the number of farmers willing and able to adopt these systems. Within this analysis, time-related costs were assessed qualitatively based on preferences expressed through interviews.

The results of this study could provide value to farmers and other stakeholders in the EA irrigation market in several key ways. The sets of irrigation product requirements for each market segment could help irrigation equipment designers target innovation efforts. Irrigation companies and NGOs could benefit from understanding the unique needs of new markets identified. Farmers in new and existing markets could benefit from irrigation products tailored to their needs, increasing their likelihood of adoption. Increased adoption and use of irrigation products could contribute to the growing need for food production in EA. The process and methodology created here could be valuable for researchers working in similar regions or designers working in global contexts to address the needs of underserved markets. In those contexts, this process can be repeated to identify new relevant market segments and elucidate areas for further innovation.

## 5. Conclusions

This study aimed to elucidate key market segments and identify new market-specific opportunities for innovation that might enhance the adoption of farmer-led irrigation systems in EA. Four key market segments were identified among farmers who cultivate ≤5 ha—the traditional smallholder, the semi-commercial smallholder, the medium-scale contract farmer, and the remote farm owner. Unique needs and value propositions were created for each market segment, informed by farmer interviews, farm tours, stakeholder interviews, and the literature. A techno-economic analysis was used to estimate the costs of irrigation systems capable of meeting farmers' unique irrigation needs. These two analyses were synthesized to identify new opportunities to create irrigation systems with a high potential to increase irrigation adoption in EA. In the traditional smallholder market, this work identified the potential for a system that combines PV panels + manual irrigation. For the semi-commercial smallholder, the study identified a PV panel + butterfly sprinkler-based system. Finally, for medium-scale contract farmers and remote farm owners, the study identified a PV panel + NPC drip-based system.

The results of this study demonstrate that none of the proposed irrigation systems are estimated to be low-cost enough to meet the price constraints of small-to-medium-scale farmers in EA. However, the results identified opportunities for technical innovation to better serve each market segment. Several key opportunities were identified, such as the design of lower-cost, longer-lasting pumps; lower-cost, longer-lasting NPC or LE PC drip lines; and anti- or low-clogging NPC or LE PC drip emitters. Key innovation in these areas would help increase the adoption of drip technology in EA. For the semi-commercial smallholder, this study identified a need to create irrigation systems that allow a farmer to modularly expand their irrigated area. For example, a farmer could irrigate 0.25 ha with butterfly sprinklers or 0.5 ha with NPC drip irrigation while using the same pump and PV panels. This system architecture would allow the farmer to invest in a new irrigation method without investing in an entirely new system.

This study highlights the many opportunities to design irrigation systems that fulfill farmer values and needs beyond their irrigation requirements alone. For the traditional smallholder, this study highlighted a system that provides phone charging and home lighting in addition to irrigation. For the semi-commercial smallholder, it identified an opportunity to create a system that includes small home appliances. For the medium-scale contract farmer and remote farm owner, the study found that each would benefit from low-cost data and prediction tools that support their farm management. The remote farm owner would also benefit from tools that help them ensure the quality of labor and care for their crops when they are off-site. The needs of this emerging market segment will likely evolve as the market matures. Future work will explore innovations to address some of the key opportunities identified here. Farmers and other market stakeholders will continue to be engaged to refine user needs as prototypes of the innovations are built and tested.

**Author Contributions:** Conceptualization, G.D.V.d.Z., S.A. and A.G.W.V.; methodology, G.D.V.d.Z. and E.D.; formal analysis, G.D.V.d.Z.; investigation, G.D.V.d.Z., E.D., S.A. and P.S.; data curation, G.D.V.d.Z., E.D. and P.S.; writing—original draft preparation, G.D.V.d.Z.; writing—review and editing, all authors; funding acquisition, S.A. and A.G.W.V. All authors have read and agreed to the published version of the manuscript.

**Funding:** This study was supported by a gift from the Julia Burke Foundation.

**Institutional Review Board Statement:** All interview protocols were approved by the Massachusetts Institute of Technology Committee on the Use of Humans as Experimental Subjects (protocol numbers E-1508 and E-1990).

**Informed Consent Statement:** Informed consent was obtained from all subjects involved in the study.

**Data Availability Statement:** The data presented in this study are available on request from the corresponding author. The data are not publicly available due to this set of human subject data is not publicly shareable.

**Acknowledgments:** The authors thank the Julia Burke Foundation for supporting this work. They would also like to thank all interviewees—farmers and market stakeholders—for sharing their insights on the EA irrigation market. Finally, they thank the organizations and companies who connected them to interviewees: SunCulture, Global Good, Futurepump, Water4, Illumina Africa, Alliance Bioversity International, and iDE.

**Conflicts of Interest:** The authors declare no conflict of interest.

## Appendix A. Details of Market Segment Profiles

This appendix gives details that were used to build the farmer market segment profiles shown in Sections 2.2.1–2.2.4.

### *Appendix A.1. Traditional Smallholder*

Responses of farmers interviewed revealed that the main farming motivation for traditional smallholders is to grow food for their families. All farmers interviewed in this

market segment consumed the majority of the food they produced. Of the 10 traditional smallholders interviewed, all farmers sold only a small portion of their produce in local markets. Instead, these subsistence-focused farmers primarily produced food for in-home consumption. As a result, irrigated areas in this market segment tended to be the smallest. A typical irrigated area among this segment was 0.125 ha, though non-irrigated cultivated land and non-cultivated holdings were typically larger. In an interview, members of the One Acre Fund, an NGO that provides input services to smallholders in EA, corroborated that this amount of land alone could sustain an average family. Farmers reported growing at least four crops to meet their family's dietary needs. These crops varied by region and included maize, cassava, teff, cabbage, onion, and kale. In addition, many farmers reported a desire to grow a wide variety of crops in the future, demonstrating the value they placed on crop diversity.

Traditional smallholders tended to have experience with only one irrigation method, the one they were using at the time of the interview. However, the low penetration of irrigation among farmers with <2 ha (0.7–2.3% across EA) suggests that even this minimal experience is above average and that the vast majority of traditional smallholders have minimal or no irrigation experience [45]. This lack of irrigation penetration may be related to the remoteness of these farmers, all of whom lived far from cities. Agricultural innovation has been found to be lower in remote regions where disseminating agricultural information is difficult [13]. This barrier is consistent with interview responses, which confirmed that both information access and training opportunities were limited among traditional smallholders.

Of the EA traditional smallholders who do irrigate, most rely on manual irrigation [7]. This is known to be labor-intensive and many farmers reported having to carry water from a distant water source to their fields. In Zambia, where Water4 (an NGO) recently installed solar-powered submersible pumps on local farms, one farmer reported that his previous manual irrigation practice required over four hours each day to fill a 630 L tank from a lake source 300 m away. This experience was typical among smallholders who previously relied on manual irrigation. Other farmers reported that manual irrigation was not only tiring but also dangerous. Several farmers in this area raised concerns about crocodiles attacking humans during water collection, reporting that at least one farmer per year is killed by crocodiles while fetching water. While interviewed farmers no longer faced these issues, these experiences are likely to be typical of smallholders in the region who continue to use manual irrigation.

Attitudes towards manually powered pumps with low capital costs, such as the treadle pump, revealed both the high value placed on low-cost irrigation and the high physical toll of supplying the water manually. A 66-year-old farmer in Ethiopia reported operating his treadle pump for at least an hour each day and feeling exhausted by the effort. In interviews, several high-level executives at a company that sells treadle pumps also remarked how tiring their pump is to use. These results suggest that a large value added to these farmers' daily lives would be an irrigation system that does not rely on human power. Some traditional smallholders use small fuel pumps, and they expressed how valuable it was to not have to carry water.

Attitudes towards risk and income generation patterns suggest that traditional smallholders tend to be very risk-averse and would value a system that they know they could pay for in 2–3 seasons' worth of profits. These smallholders use income source diversification as a risk management strategy [18]. For EA countries, the Food and Agriculture Organization of the United Nations reports that 43–62% of small family farm income comes from crop production while the remainder comes from other sources, suggesting that they are not willing to rely solely on high-risk agriculture for income [45]. This was consistent with behaviors observed throughout the interviews. In total, 4 of the 10 interviewed traditional smallholders noted additional sources. This diversification of income suggests an increased risk aversion and an increased need for a financial guarantee. Traditional smallholders think about their farming future 2–3 seasons out. This short payback time frame reported

by farmers suggests that they want to have a guarantee that an irrigation system would be paid back in that time period. If not, they may risk diving deeper into poverty. Multiple farmers who were still paying back a system expressed deep concern over potentially defaulting on loans. Farmers are not the only stakeholders who want to see a quick return on their investments. In interviews, NGOs that provide loans to traditional smallholders stressed the need for farmers to be able to pay for a system in less than one year because they do not trust farmers to pay back longer loans. The interviewed MFIs do not currently provide loans to traditional smallholders for irrigation equipment because it is too risky for the MFI. They have in the past, but too many farmers defaulted, leaving the MFIs to refuse new farmers who fit this profile. Many farmers in this segment are below or just above the poverty line [18,46], and their pattern of income diversification suggests that they are unlikely to invest all their additional income or savings into one system, even if that system increases their agricultural productivity. A high-performing, inexpensive system that can be paid back in 2–3 seasons would provide value to both traditional smallholders and the stakeholders who serve them.

Increasingly more farmers in this segment have home lighting [27] and cell phones [28], two products the interviewed farmers valued if they had access. One traditional smallholder in Ethiopia commented on a standalone lighting product he owned. He paid 3500 birr (USD 122) for it and believed it was worth the cost; for reference, a treadle pump costs USD 170 [47]. Current irrigation systems that serve traditional smallholders do not incorporate USB ports for these valuable features, but our results suggest that the majority of traditional smallholders would value this add-on.

*Appendix A.2. Semi-Commercial Smallholder*

The semi-commercial smallholder was likely a traditional smallholder at one time. Now, they have moved away from subsistence farming, seeing how they can start a small farming business. Compared to the traditional smallholder, they are more willing to invest both time and money in equipment that has a promising return on investment because they have seen past success in agriculture. In total, 13 of the 14 interviewed semi-commercial smallholders used PV-powered irrigation systems sold by Futurepump or SunCulture. This indicates farmers' willingness to invest in more expensive equipment. Depending on the specifications and added features, these systems cost between USD 600 and 1550, which farmers pay for over 2–3 years [30,48].

Compared to traditional smallholders who have diverse income sources, farmers in this market segment are more focused on farming as their main income source. Therefore, they are able to dedicate more irrigation time per day than traditional smallholders can. Five of the interviewed farmers spent at least 4 h irrigating each day, with three of them spending over 8 h. Of the nine who spent less than 4 h irrigating per day, seven used NPC drip irrigation. They spent 1–2 h monitoring the irrigation, and then they could let it run while they focused on other tasks, meaning the farms were being irrigated for longer than 4 h/day. It is estimated that semi-commercial smallholders will spend up to 6 h/day irrigating, especially if they do not need to continuously monitor it.

Not all semi-commercial smallholders were once traditional smallholders. City dwellers who move to the country for retirement can also fit this profile. This was the case for 3 of the 14 interviewed farmers. Their motivations were to sustain their own diets and to supplement their retirement funds by selling the remaining produce. In these cases, the farmers were still confident they could make their monthly payments even if they had a few unsuccessful seasons.

Interviewed semi-commercial smallholders grew largely the same types of crops as traditional smallholders, with a slightly higher focus on fruits and vegetables over grains. Fruits and vegetables are all higher value crops than grains like maize, teff, or cassava [49]. Given commonalities between all interviews, it is estimated that semi-commercial smallholders sell 30% of their product in a local market or to a middleman who transports it to a nearby city. One farmer who has been farming for almost 25 years did not sell any product

until he purchased one of SunCulture's PV-powered systems: the RainMaker2. Now, he estimates he sells between 50 and 100 kg of produce each week during the harvest season.

Semi-commercial smallholders are quick to implement new agriculture techniques when they have access to the right resources. The majority of interviewed farmers in this segment had access to some form of professional training. Certain RainMaker2 models have a television bundled into the irrigation product. This television comes preloaded with Shamba Shape Up, a TV series that teaches improved farming techniques [50]. One farmer said this content was his favorite part of his RainMaker2 product. After watching the tutorials, he was able to confidently raise chickens to expand his agriculture business. Other farmers were trained to use their irrigation equipment by representatives from the distributor. One farmer said she received about an hour of training on her Futurepump PV-powered irrigation system when it was installed. Another farmer had inexpensive, non-pressure-compensating (NPC) drip irrigation lines installed in his greenhouse. He was trained on how to use them for daily use but not how to flush them to prevent emitter clogging. Two farmers mentioned that they were curious about drip irrigation, but did not yet know how to use it. While there is a need for better training in this market, the current training that these farmers receive is better than what the traditional smallholders typically have access to. This means they are able to adopt more advanced irrigation methods, like drip irrigation.

Farmers in this segment derive value from small home appliances, like televisions and pressure cookers, and they are willing to pay for these items. Pressure cookers allowed farmers to cook warm food faster than their previous methods. Televisions gave farmers a source of entertainment in addition to the Shamba Shape Up tutorials. In interviews, SunCulture leadership stressed the success of bundling a television with their irrigation systems. At the time of the interview, their highest-selling product was the RainMaker2 with ClimateSmart™ Battery + TV. These systems had such high sales that SunCulture has since started selling a system with just the battery, television, and home lighting: the ClimateSmart™ Battery + TV [30]. In interviews, farmers asked for agricultural products that could pair with their systems as well. A popular request was a chaff cutter, followed by an egg incubator. This suggests that these types of appliances, in addition to home lighting and phone charging, would increase the likelihood that semi-commercial smallholders adopt an irrigation system.

*Appendix A.3. Medium-Scale Contract Farmer*

Medium-scale contract farmers run full-time farming businesses to feed the growing cities in EA [51–53]. They cultivate medium-sized farms in peri-urban areas. The interviewed farmers irrigated 1.2–8 ha, but they owned more, between 2.4 and 20 ha, suggesting there is an opportunity to expand irrigation on these farms. In support, one stakeholder who had previously served contract farmers claimed there were still many underserved farmers who owned 2–6 ha. To supply food to nearby cities, these farmers have contracts with middlemen who deliver their produce to urban supermarkets, hotels, universities, or airlines, for example. Farms are within a few hours' drive to these destinations, so there is a chance they have grid connections. However, these connections may not be reliable as outages are common [40].

Farmers in this market segment invest in their businesses. Intending to sell >95% of their produce, they cultivate high-value crops like tomatoes, herbs, and fruit. Two farmers reported the costs of their irrigation systems: USD 18,500 for a system that irrigates 8 ha and USD 30,000 for a system that irrigates 1.2 ha. This second system cost includes the cost of drilling a 300 m deep borehole. Farmers in this segment invest in equipment, planning on a 5–10-year timeline.

Medium-scale contract farmers employ seasonal and full-time laborers who irrigate, weed, plant, and harvest. Because farmers have this additional help, they are willing to spend the whole solar day irrigating, an estimated 7 h. Five of seven interviewed farmers said their irrigation systems run for longer than 5 h each day.

Medium-scale contract farmers have advanced irrigation experience compared to the smallholders, but they still experience challenges. Five of the seven interviewed farmers used NPC drip irrigation, and all of them were familiar with the technology. Farmers liked that drip irrigation let them irrigate without much oversight. They or a laborer could open one section of the network and then perform non-irrigation tasks for 30–60 min until they needed to switch to another section. Stakeholders confirmed this benefit, but noted that that drip only works well for farmers who have learned how to use drip effectively. Farmers and stakeholders alike noted that emitter clogging was a large drawback of this technology. Sediment can collect in the small emitter features, blocking the flow of water. Farmers need to follow proper filtration and flushing regiments to avoid this, but not all do.

Because these farmers focus on selling their produce, the appearance and size uniformity of their crop is important. Pressure-compensating (PC) drip is typically preferred for increased crop uniformity because it regulates the flow of all drip emitters in a network, but stakeholders who design irrigation systems for this market segment said they always recommend NPC drip over PC. They do recommend PC drip to floriculturists who have even higher uniformity standards. However, for medium-scale contract farmers, system designers do not see how the added value of PC drip outweighs the higher equipment cost. To overcome the uniformity drawbacks of NPC drip, irrigation systems are designed with small sections (about 0.2 ha), with laterals no longer than 30 m. For comparison, in regions that have higher PC emitter adoption rates, like India, laterals can be up to 75 m long. Current 0.2 ha sections are irrigated for only 30 min at a time, which means section valves are turned on and off frequently. Longer laterals could reduce labor needs as irrigation sections could be larger. Having larger sections means having fewer sections and fewer people monitoring the irrigation schedule changes.

*Appendix A.4. Remote Farm Owner*

The remote farm owner lives in a city but owns or rents land in a nearby peri-urban region. They farm as a hobby or as a way to make supplemental income while investing in the land. While the remote farm owner may be involved in making big decisions about the farm, they are not present on a daily basis. Instead, they hire farm managers and laborers to run the farm for them. One Nairobi-based remote farm owner visits his farm almost every weekend and pays 4–5 laborers to tend during the week.

The remote farming market segment is an emerging one and not all problems with managing a farm remotely have been solved, so there is risk involved for the owners. For example, one stakeholder who sells seedlings to farmers has had several remote farming customers. He recommends they avoid this model, predicting they will "be taken for a ride". For example, one of his customers bought chemicals for their farm. Their laborers claimed to have sprayed them, but in reality, the chemicals were resold. The quality of the crops was evidence that the plants did not receive appropriate care. The Nairobi-based remote farm owner agreed that his laborers do not show the same quality of work when he is not on site.

Farmers in this segment have the capital to invest in irrigation systems, but they do not intend for farming to be their main income source. A second interviewed remote farm owner has been running a real estate company for 10 years. He had no knowledge of farming but saw this as a growing business opportunity. At the time of the interview, he was still in the process of setting up the farm, but he had high hopes based on the success he has seen with other remote farm owners in his network.

**Appendix B. Elaboration on Irrigation Method and Energy Source Parameters in Tables 3 and 4**

*Appendix B.1. Irrigation Methods*

Appendix B.1.1. Manual Irrigation

Manual irrigation, using buckets or handheld hoses to deliver water to a field, is one of the most commonly used irrigation methods among EA smallholders [3]. One reason for this popularity is its low cost, an estimated capital cost of USD 10 that needs replacement every two years [54]. While inexpensive, manual irrigation is very labor-intensive, limiting the area that can be irrigated in a single day by a single farmer. Farmers can only irrigate one plant at a time, so the estimated maximum area they can irrigate in a day is 0.2 ha. Assuming no additional labor is hired, these estimations give manual irrigation an equipment cost of 50 USD/ha.

Manual irrigation can deliver water-usage benefits because farmers walk with and constantly monitor their irrigation amount. Farmers are likely to only water the base of the crops and not the space in between, so the estimated water factor for manual irrigation is 0.5. Manual irrigation takes very little pressure head to operate, an estimated maximum of 1 m [55]. If a bucket is used, the head needed is 0 m. If a hose is used, no more than 1 m of head is needed.

A key drawback of manual irrigation is that it is a physically demanding job for farmers as they must continually walk with the equipment. If they are using buckets, they are also tasked with carrying the water load. This drawback is analyzed further when discussing results from the farmer interviews.

Appendix B.1.2. Flood and Furrow Irrigation

Flood irrigation, covering the entire field with water, or furrow irrigation, filling furrows between crop beds with water, are two traditional irrigation methods commonly used by smallholders and medium-scale farmers in EA [56]. These irrigation methods use very little equipment, so the low cost makes them a popular option. In addition to a pump, only a 50 m hose pipe is needed to direct the water flow, and there is an estimated cost of 25 USD/ha with a lifetime of two years [54]. This hose pipe would not need more than 5 m of pressure head.

A big drawback to flood and furrow irrigation is the high water usage, with an estimated water factor of 1.0 [57]. The entire field must be covered with water for flood irrigation, and about half of the field area is covered for furrow irrigation. Still, the furrows must be filled with water, so overwatering is common, bringing the estimated water factor to 1.0 for both methods. According to a key stakeholder who sells irrigation products to farmers, a second drawback to flood or furrow irrigation is the significant amount of skill needed to prepare the field, which could deter farmers with little irrigation experience.

Appendix B.1.3. Butterfly Sprinklers

Butterfly sprinklers are increasing in popularity among smallholders in EA, and they are included in irrigation kits like SunCulture's RainMaker2 series [30]. For this analysis, it is assumed that a farmer uses one set of five sprinklers that they move throughout their field. In this setup, a farmer moves the sprinkler heads every 30 to 60 min.

This butterfly sprinkler set operates at 10 m of pressure head and costs an estimated USD 26.50, with a lifetime of two years. According to irrigation system designers, butterfly sprinklers cost USD 0.30 per unit. A 50 m hose to connect the sprinklers to a pump costs USD 25 [54]. For this estimation, it is assumed that the five sprinklers operate in parallel and that one sprinkler plus one 50 m hose operate in a range centered around 10 m of pressure head [58]. Because they are operated in parallel, the entire set of five sprinklers also operates at 10 m of pressure head.

Two drawbacks of sprinklers are their high water usage and their need for farmer labor. Sprinklers distribute water to the entire field with a water factor of 1.0 [57]. In the assumed set of five movable sprinklers, there is labor involved, but less than in manual irrigation.

Appendix B.1.4. NPC Inline Drip Irrigation

According to a stakeholder who owns an irrigation equipment company in Kenya, NPC inline drip irrigation is commonly used by farmers in EA who focus on selling a large volume of crops. Drip irrigation delivers water to rows of crops through a network of stationary main and submain pipes and lateral lines. At the base of each plant, an emitter bonded to the inside of the lateral line allows water to flow. Because drip only delivers water to the root bases, the estimated water factor is 0.5 [57].

Netafim's Streamline™ X, a popular emitter globally, is used as a representative NPC emitter that operates at a range centered around a 10.2 m pressure and a 2.2 L/h flow rate [59]. Non-pressure-compensating drip emitters do not regulate the flow of water when varying pressure is applied. This means that, on a flat field, emitters at the end of a lateral line will be lower in flow than emitters at the beginning of that line. It also means that a graded field will result in non-uniform flow rates.

A 0.2 ha section of NPC drip operates at 14 m of pressure head and costs 2400 USD/ha in EA with an equipment lifetime of three years. Stakeholder interviews suggest that typical NPC drip sections in Kenya were 0.2 ha, or 30 m by 67 m. They have 100 30 m long lateral lines with 0.3 m crop spacings. The operation of this 0.2 ha pipe network was simulated using a systems-level model [60,61], showing that it works at a pressure head of 14 m. Interviews with stakeholders showed that NPC drip irrigation (including lateral lines, emitters, submain and main pipes, and valves) in Kenya costs an estimated 2400 USD/ha. It is expected that this equipment has a lifetime of three years before the plastic lines degrade or crack [59].

Appendix B.1.5. LE PC Inline Drip Irrigation

LE PC inline drip irrigation is a method that is not yet widely used but shows promise in the region [39]. It relies on novel emitters designed by the MIT GEAR Lab that have the potential to save 42–54% in pumping power over standard-pressure emitters for surface-water systems [38,62]. Unlike NPC emitters, PC emitters use silicone membranes to regulate water flow once an activation pressure is reached. The consequence of this flow rate regulation is increased flow uniformity throughout the field, which leads to increased crop size uniformity. The GEAR Lab has recently developed LE PC emitters that activate at 1.5 m head with 2.2 L/h flow rates. Compared to conventional PC emitters like Netafim's UniRam™ RC (Hatzerim, Israel) that activate at 5.0 m [63], LE PC emitters show promise for reducing the capital cost of solar-powered systems or the operating costs of the grid- and fuel-based systems [38].

A 0.2 ha section of LE PC drip operates at 5.9 m of pressure head and costs 6000 USD/ha with a lifetime of ten years. LE PC drip is expected to cost a similar amount as conventional PC drip costs, 6000 USD/ha. One interviewed stakeholder who sells drip equipment estimates that PC lines cost roughly 2.5 times more than NPC in Kenya. This discrepancy is due to the thicker pipe walls of PC lines and the added silicone membranes in each emitter. LE PC drip equipment is expected to have a similar lifetime as conventional PC drip: ten years [63]. Like NPC drip, the water factor of LE PC drip is 0.5.

*Appendix B.2. Energy Sources*

Appendix B.2.1. PV Panels

PV panels are becoming an increasingly popular energy source across all EA market segments as panel prices decrease [44]. The 2019 Global LEAP Awards estimates the price of solar panels in EA to be 0.81 USD/W [64]. This cost estimate was confirmed as a reasonable estimate with stakeholders at Illumina Africa, a PV panel installation company based in Nairobi, Kenya. The lifetime of a solar panel is estimated to be 20 years [65].

Solar panels are viable options for many regions in EA, rural or peri-urban, because EA records high solar irradiances for the majority of the year [12]. PV panels are one of the most sustainable energy sources, so they are being promoted by NGOs, companies, and governments. For example, import taxes are waived on PV panels used for agriculture in Kenya [66].

### Appendix B.2.2. Grid Electricity

Grid electricity is an option for some EA farms that are close to cities and have existing grid connections. In Kenya in 2015, 17.1% of rural households and 73.0% of urban households had grid connections, suggesting that farmers in peri-urban areas may also have access to grid electricity [67]. Electricity costs in Kenya are 0.06 USD/MJ [68]. Installing new grid connections can be prohibitively expensive to farmers in EA [40], so it was outside the scope of this study to consider new connections.

### Appendix B.2.3. Fuel

Fuel is currently the most popular external energy source for irrigation among smallholders in EA [69]. Fuel prices fluctuate by time and region, but are estimated to be 0.03 USD/MJ, from 1.025 USD/L of diesel [70] and an energy density of 36 MJ/L. Capital and replacement equipment costs are not applicable for grid or fuel energy sources because the recurring cost of purchasing energy dominates any estimation.

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
