# Peer review of "Identifying Opportunities for Irrigation Systems to Meet the Specific Needs of Farmers in East Africa"

_water, doi:10.3390/w16010075_

Round 1

Reviewer 1 Report

Comments and Suggestions for Authors

This study has identified four distinct market segments: traditional smallholders preferring photovoltaic (PV) and manual irrigation, semi-commercial smallholders favouring PV panels and butterfly sprinklers, and medium-scale contract farmers along with remote farm owners valuing PV panel- and drip irrigation-based systems. The study emphasizes that targeting these specific preferences with innovative irrigation solutions could significantly contribute to addressing food shortages and increasing food security in the EA region. Identifying the most effective irrigation system based on the farmers’ perspective is an important and interesting study area. The authors have effectively developed the methodology to obtain relevant data in each sector, addressing the research problem. Overall, the paper is well-structured.

While the study provides valuable insights into market segments and their preferences for irrigation systems in East Africa, I would like to highlight some potential areas to improve the paper's findings:

1.      1.  Study area/location: Include a clear location map of the study area to visually recognize the study location/s.

2.      2.  In line 374 - fw1: Change the notation according to Equation 1.

3.      3. Figure 2: The use of a sprinkler system may cause more water loss (e.g., due to evaporation, etc.) in comparison to drip irrigation; hence, there may be a need for more operating time. Have you considered this into account/will there be any significant cost change?

4.      4.  Further, did you consider soil condition in the study area and water use efficiency over furrow, sprinkler, and drip irrigation systems? This, again, may affect the operating time of equipment.

5.       5. The study has conducted a technical and economic feasibility analysis of the selected systems; however, this could include a detailed cost-benefit analysis and considerations of affordability for different farmer segments, which may provide a different perspective and may help to change the farmers’ insight.

Reviewer 2 Report

Comments and Suggestions for Authors

The article "Identifying opportunities for irrigation systems to meet the specific needs of farmers in East Africa" presents a comprehensive attempt to understand the challenges faced in increasing irrigation adoption among farmers in East Africa. While the study offers valuable insights, there are certain weaknesses and areas that could be improved:

The article introduction needs more improvement with proper citations. Kindly add more rationale related to the study.

  1. The article mentions the diversity among East African farmers but falls short in providing a detailed breakdown of these demographics. A more granular analysis of factors such as geographic location, socio-economic status, and specific farming practices would have added depth to understanding the unique needs of each segment.

  2. The interview-based market assessment, while a good starting point, might have missed out on a broader representation of farmers. A larger sample size and a more diverse range of interviewees across different regions and farming practices would have yielded a more comprehensive understanding.

  3. The technical and economic feasibility analysis seems to assume homogeneity within each identified segment. It's essential to acknowledge that within each segment, there can be variations in resource availability, land size, and technological literacy, which could impact the feasibility of suggested irrigation solutions.

  4. The study primarily focuses on technical and economic aspects without delving deeper into the environmental impacts and sustainability of the proposed irrigation solutions. Factors like water conservation, ecological footprint, and long-term sustainability should be incorporated into the analysis.

  5. While the article identifies opportunities for innovation in irrigation systems, it lacks a forward-looking approach in discussing potential challenges or technological advancements that might affect these opportunities in the future.

  6. The role of governmental policies, institutional support, and infrastructural limitations in hindering or facilitating irrigation adoption among East African farmers is not extensively discussed.

Comments on the Quality of English Language

Minor editing of English language required.

Reviewer 3 Report

Comments and Suggestions for Authors

This is the opinion of the article "Identifying opportunities for irrigation systems to meet the specific needs of farmers in East Africa". 

The article sought to carry out a very interesting study. 

Some adjustments need to be made, as follows:

1) better contextualize the problem and the hypothesis. 

2) adjust the key words, in alphabetical order, and include words and expressions.

#) the text needs to be revised, there are parts that are a bit confusing. 

Comments on the Quality of English Language

This is the opinion of the article "Identifying opportunities for irrigation systems to meet the specific needs of farmers in East Africa". 

The article sought to carry out a very interesting study. 

Some adjustments need to be made, as follows:

1) better contextualize the problem and the hypothesis. 

2) adjust the key words, in alphabetical order, and include words and expressions.

#) the text needs to be revised, there are parts that are a bit confusing. 
